# Covalently linked molecular catalysts in conjugated polymer dots boost photocatalytic alcohol oxidation in neutral condition

Sicong Wang[1], Mariia V. Pavliuk[1], Xianshao Zou[2], Ping Huang [1], Bin Cai[1], Orpita M. Svensson[1] & Haining Tian [1] ✉

As a new class of organic photocatalysts, polymer dots show a potential application in photocatalytic hydrogen peroxide production coupled with chemical oxidation such as methanol oxidation. However, the poor methanol oxidation ability by polymer dots still inhibits the overall photocatalytic reaction occurring in the neutral condition. In this work, an organic molecular catalyst 4-amino-2,2,6,6-tetramethylpiperidine-1-oxyl radical is covalently linked to a fluorene unit in a polymer skeleton, eventually enabling photo-catalytic hydrogen peroxide production coupled with methanol oxidation in the neutral condition. By conducting various spectroscopic measurements, charge transfer between components in this molecular catalyst-immobilized polymer dots system is studied and found to be very efficient for hydrogen peroxide production coupled with alcohol oxidation. This work proves a strategy for designing polymer dots photocatalysts with molecular catalysts, facilitating their future development and potential applications in other fields such as water splitting, $CO_2$ reduction, photoredox catalysis and photo-dynamic therapy.

As a widely-used industrial product (e.g. textile, food, mining and pulp), hydrogen peroxide ($H_2O_2$) has shown increased demand by over 260% from 1.5 million tons in mid-1990 to 5.5 million tons in 2015[1,2]. However, the anthraquinone method which dominates 98% of industrial $H_2O_2$ production produces substantial waste and will lead to a more negative environmental impact due to the rapid increase in global demand for $H_2O_2$[3,4]. Photocatalytic $H_2O_2$ production through oxygen reduction reaction (ORR) has been considered as a promising alternative method and therefore has emerged as a global subject of research in recent decades[5,6].

Owing to advantages such as non-metal properties and tunable energy structures, organic photocatalysts have been intensively investigated for photocatalytic production of $H_2O_2$ through ORR

mechanisms, such as graphitic carbon nitride[6–8], covalent organic frameworks[9] and conjugated polymers[10,11]. Among conjugated polymer photocatalysts, polymer dots (Pdots), as a kind of polymer nanoparticles (NPs) with particle size less than 100 nm, have shown good performance in photocatalysis due to large surface area and efficient charge separation[12]. Pdots can also be prepared into hetero-junction compositions, for example, consisting of an electron-donating polymer (donor) and an electron-accepting molecule (acceptor) in which the effective interaction between the donor and acceptor can facilitate fast charge transfer (femtosecond-picosecond level)[13–16]. Nonetheless, a typical photocatalytic $H_2O_2$ production through the ORR process has limits such as fast charge recombination, slow kinetics and possible decomposition of generated $H_2O_2$[17,18].

[1]Department of Chemistry - Ångström Laboratory, Uppsala University, 751 20 Uppsala, Sweden. [2]Qingdao Innovation and Development Base, Harbin Engineering University, Qingdao CN-266 000, China. ✉e-mail: haining.tian@kemi.uu.se

Therefore, producing $H_2O_2$ coupled with other value-added organic products (e.g. formic acid) is also meaningful since photogenerated holes can be efficiently utilized by proceeding oxidation reactions[19,20].

Various strategies have been applied in polymer NPs systems to achieve promising photocatalytic proton and oxygen reduction activities, such as morphology controlling[15,16], Au nanoparticles loading[21], polymer side chain adjustment[22], topological structure engineering[23,24] and donor-acceptor unit designing[25]. However, grafting molecular catalysts onto a polymer skeleton, which is widely used in bulk organic photocatalyst systems, has not been studied in Pdots systems for $H_2O_2$ coupled with chemical oxidation so far[26–33]. Recently our group reported Pdots consisting of Poly(9,9-dioctyl-fluorene-alt-benzothiadiazole) (PFBT) as donor and 1-[3-(Methoxycarbonyl)propyl]–1-phenyl-[6.6]$C_{61}$ (PCBM) as acceptor that showed a promising photocatalytic $H_2O_2$ and formate production rate of 188 mmol h$^{-1}$ g$^{-1}$_{Pdots} in alkaline conditions through ORR and methanol (MeOH) oxidation[34]. However, when the system was placed in the neutral condition, the $H_2O_2$ production was severely inhibited due to the difficult MeOH oxidation by PFBT[34]. This therefore encouraged us to develop a Pdots system that can accelerate the oxidation of MeOH and then produce $H_2O_2$ in the neutral condition, because the production of $H_2O_2$ in the neutral condition has broader applications such as pollutant treatment[35,36] as well as photodynamic therapy[37,38]. To accomplish MeOH oxidation in the neutral condition, 2,2,6,6-Tetramethylpiperidine-1-oxyl radical (TEMPO) as an organic molecular catalyst for alcohol oxidation[39,40] is selected to decorate the PFBT backbone in this work.

In this study, TEMPO covalently immobilized and carboxyl groups functionalized PFBT polymer, coded as PFBT-T and PFBT-COOH, respectively, were systematically studied in photocatalytic hydrogen peroxide formation coupled with MeOH oxidation in the neutral condition. PCBM was utilized as a molecular electron acceptor to prepare binary Pdots (PFBT-T/PCBM) with PFBT-T. As compared to the previously reported PFBT/PCBM Pdots[34], the binary PFBT-T/PCBM Pdots showed ground-breaking production rates of both $H_2O_2$ and formaldehyde of 865 μmol h$^{-1}$ g$^{-1}$_{Pdots} and an external quantum yield (EQE) of 0.75% at 450 nm in the neutral condition. The system has also shown general application in the oxidation of other alcohols such as benzyl alcohol and ethanol.

## Results

The synthetic route of PFBT-T is shown in Fig. 1a. PFBT-COOH was synthesized via the Suzuki coupling reaction. Subsequently, 4-amino-2,2,6,6-tetramethylpiperidine-1-oxyl (TEMPO-NH$_2$) was covalently grafted to PFBT-COOH polymer backbone by a two-step synthesis. The molecular weight of PFBT-COOH was determined to be 2300, which means that the polymer has 5 fluorene and 5 benzothiadiazole units repeated in the structure. To prove the successful grafting of the TEMPO moieties in the PFBT backbone, Fourier-transform infrared spectroscopy (FTIR) (Fig. 1b), electron paramagnetic resonance (EPR) (Fig. 1c) and cyclic voltammetry (CV) (Fig. 1d) were employed. Compared to the FTIR spectrum of commercial PFBT, PFBT-COOH shows a new peak from 1680 to 1750 cm$^{-1}$ which is attributed to the C = O stretching vibration from the modified carboxyl groups. After the linkage of TEMPO, C-N stretching from 1000 to 1200 cm$^{-1}$ was detected, indicating the existence of TEMPO moieties in PFBT-T. EPR spectra recorded for PFBT-T Pdots and PFBT-T in THF showed typically three-line features as a result of hyperfine coupling with the $^{14}$N (I = 1) of the nitroxyl radical (Fig. 1c). PFBT-T in Pdots generated an isotropic spectrum while the PFBT-T in THF generated an EPR spectrum characterized as anisotropic, as seen for the wing line at the high field side is broadened. This anisotropy with close similarity has been previously

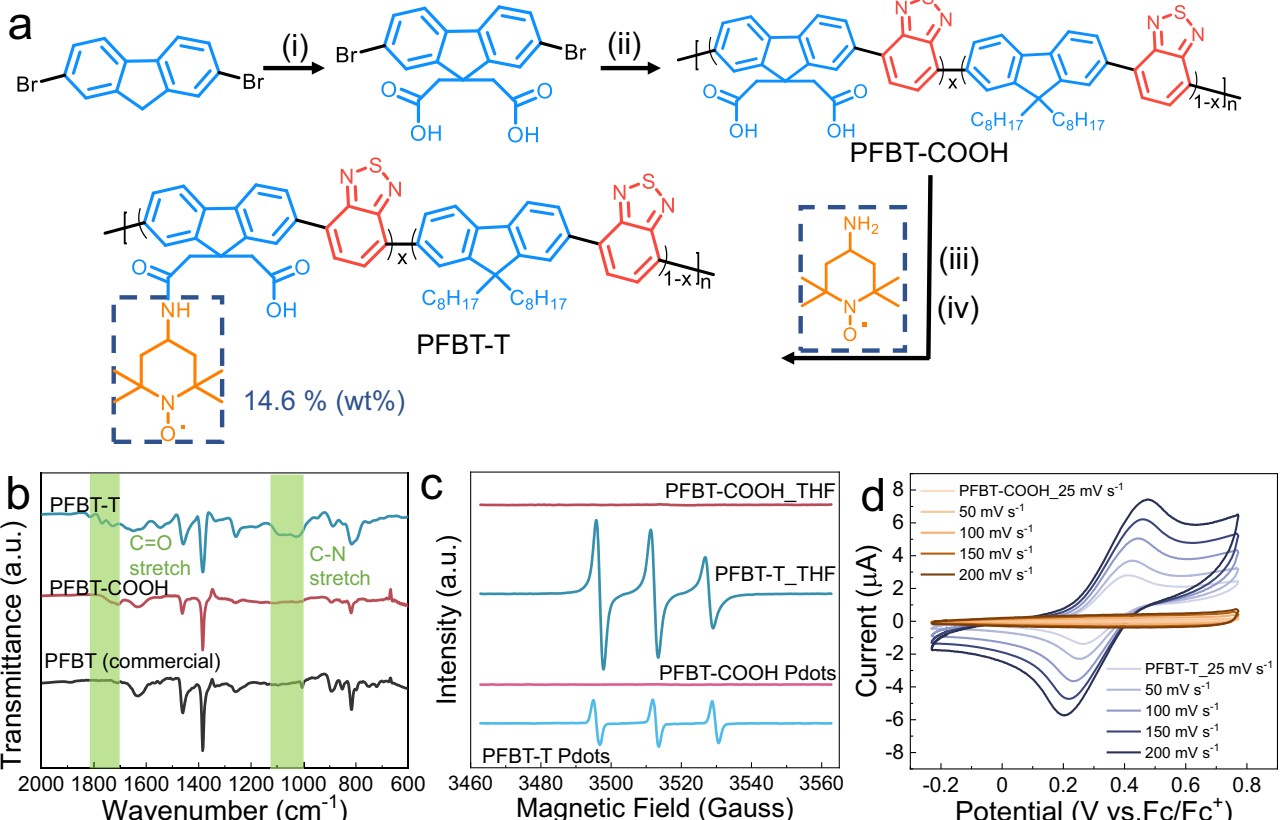

**Fig. 1 | Polymer synthesis and characterizations. a** Synthetic route of PFBT-COOH and PFBT-T, (i) ethyl bromoacetate, NaOH and DMSO; (ii) DMF, K$_2$CO$_3$, Pd(PPh$_3$)$_4$; (iii) SOCl$_2$ and reflux; (iv) TEMPO-NH$_2$ and triethylamine. **b** FTIR spectra of PFBT-COOH, PFBT-T and the commercial PFBT. **c** EPR spectra of PFBT-COOH and PFBT-T in THF and Pdots. **d** Cyclic voltammetry of PFBT-COOH and PFBT-T in THF (0.1 M TBAPF$_6$) with scan rates from 25 to 200 mV s$^{-1}$.

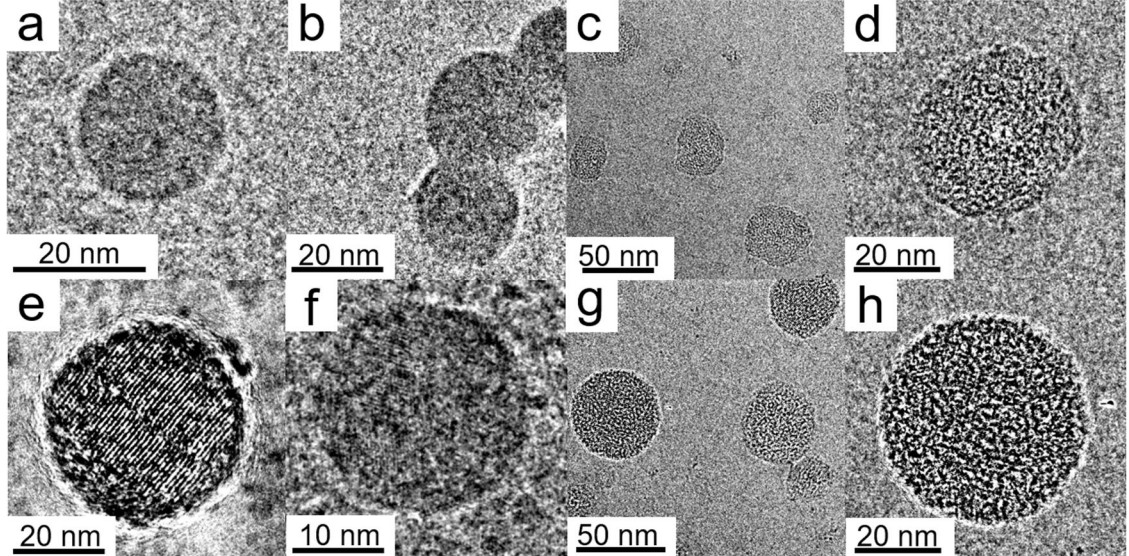

**Fig. 2 | Morphology study of Pdots.** Cryo-EM of (**a**, **b**) PFBT-COOH Pdots, (**c**, **d**) PFBT-T Pdots, (**e**) PCBM dots, (**f**) PFBT-COOH/PCBM Pdots, (**g**, **h**) PFBT-T/PCBM Pdots. Pdots and PCBM dots solution for Cryo-EM measurement are in concentrations of 1000–1200 μg mL⁻¹ after slow evaporation at room temperature.

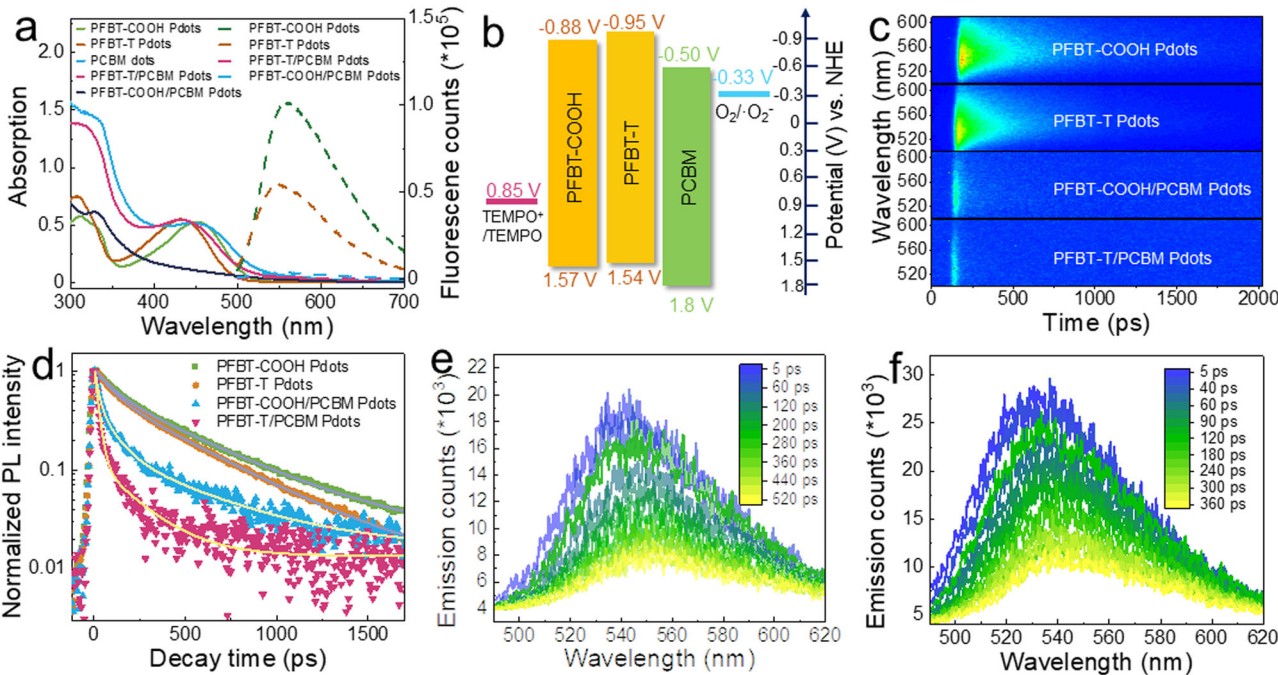

**Fig. 3 | Opitcal property study of Pdots. a** UV-Vis absorption and steady-state fluorescent emission spectra of PFBT-COOH Pdots (15 μg mL⁻¹), PFBT-T Pdots (15 μg mL⁻¹), PFBT-COOH/PCBM Pdots (28 μg mL⁻¹), PFBT-T/PCBM Pdots (29 μg mL⁻¹) and PCBM dots (17 μg mL⁻¹). **b** Energy diagram of compounds used in Pdots. **c** Streak camera spectra (excited by 400 nm laser) and **d** fluorescence decay and fitted lifetimes of PFBT* in PFBT-COOH Pdots, PFBT-T Pdots, PFBT-COOH/ PCBM Pdots and PFBT-T/PCBM Pdots. Time-dependent fluorescence of (**e**) PFBT-COOH Pdots and (**f**) PFBT-T Pdots.

reported and attributed to the tumbling correlation time of the radical[41]. If the radical is exposed to THF, as in the PFBT-T case, it is most likely forming a hydrogen bond with the solvent, which leads to a prolonged correlation time while this effect is probably being well shielded when the radical is embedded in Pdots as in the PFBT-T Pdots case. Anisotropy is averaged by the Pdots free tumbling, leading to an isotropic spectrum. In CV of PFBT-T (Fig. 1d), the oxidation of nitroxyl radicals and reduction of oxoammonium components are both clear and the oxidation and reduction currents are almost identical under different scan rates, illustrating a reversible redox process of TEMPO moieties in PFBT-T. As a comparison, no redox current is observed within the scan

range in the CV of PFBT-COOH. The physical adsorption of TEMPO-NH₂ molecules is excluded by conducting a control synthesis and relevant spectroscopic studies (see details in Supplementary Fig. 10). The above results prove that TEMPO moieties are successfully covalently linked to PFBT-T and its redox capability remains in the polymer backbone. Meanwhile, the concentration of covalently linked TEMPO is determined to be approximately 14.6% (wt.%) by conducting an EPR calibration experiment (Supplementary Fig. 11), indicating that each PFBT-T molecule contains two TEMPO units on average.

Dynamic light scattering (DLS) and Cryogenic electron microscopy (Cryo-EM) were conducted to investigate the size and

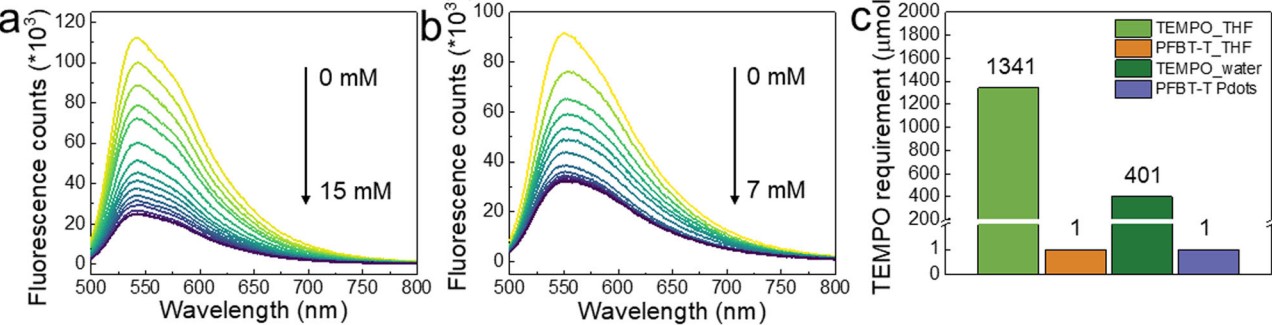

**Fig. 4 | Fluorescence quenching of TEMPO in THF and aqueous solutions.** Fluorescence quenching experiments of (**a**) PFBT-COOH (6 μg mL⁻¹) with TEMPO in THF and (**b**) PFBT-COOH Pdots (-10 μg mL⁻¹) with TEMPO in aqueous solution. **c** The required amount of TEMPO for quenching one mole of PFBT* in THF with 61% quenching efficiency and in Pdots with 47% quenching efficiency.

morphologies of Pdots used in this study. From the DLS measurement, all Pdots sizes are similar, showing a monodisperse particle distribution (Supplementary Fig. 12). From Cryo-EM images, both PFBT-COOH (Fig. 2a, b) and PFBT-T (Fig. 2c, d) Pdots have common spherical amorphous morphology which is consistent with the morphology of previously reported PFBT Pdots[13,42,43]. At the same time, PCBM dots showed clear crystalline structure characteristics (Fig. 2e)[44]. Once PCBM was blended into PFBT-COOH and PFBT-T based binary Pdots, the crystalline stripes of PCBM were found to uniformly disperse in PFBT-COOH/PCBM (Fig. 2f) and PFBT-T/PCBM Pdots (Fig. 2g, h), indicating a good intermixing with PCBM.

As seen from absorption spectra shown in Fig. 3a, PFBT-COOH absorbs up to 515 nm both in THF and Pdots. After TEMPO grafting, the absorption spectrum of PFBT-T blue-shifts about 20 nm in either THF solution (Supplementary Fig. 13) or Pdots, which can be attributed to the weakened electron-donating capability of the fluorene unit in the presence of a TEMPO group. In absorption spectra of PFBT-COOH/PCBM and PFBT-T/PCBM Pdots, a dramatic increase in absorbance at wavelengths shorter than 400 nm compared to singular Pdots is observed, which is attributed to the effective integration of PCBM into binary Pdots and is consistent with the cryo-EM results. Energy diagrams of PFBT-T, PFBT-COOH, PCBM[45], grafted TEMPO moieties[46] and ORR[47] are shown in Fig. 3b (calculation of energy diagrams of PFBT-T and PFBT-COOH is shown in SI and Supplementary Fig. 14). According to the energy levels, electron transfer from excited or reduced polymer to PCBM and hole transfer from excited or oxidized polymer to TEMPO are thermodynamically feasible.

Steady-state quenching experiments were utilized to investigate the charge and/or energy transfer from excited polymer backbone (PFBT*) to grafted TEMPO moieties and PCBM acceptor. Due to the weak absorption of PCBM dots, the hole transfer from excited PCBM (PCBM*) to PFBT is negligible under visible light irradiation and therefore is not considered. PFBT-COOH in THF shows fluorescence from 500 to 750 nm. As compared to PFBT-COOH, the fluorescence intensity is reduced by 61% in PFBT-T (Supplementary Fig. 15a). Moreover, the averaged lifetime of fluorescence is quenched from 2.2 ns in PFBT-COOH to 1.6 ns in PFBT-T (Supplementary Fig. 15b). This phenomenon is also observed in Pdots where the fluorescent emission of PFBT-T Pdots is quenched by 47% after TEMPO grafting (Fig. 3a) as compared to PFBT-COOH Pdots, proving that the PFBT* in PFBT-T can be reductively quenched by the grafted TEMPO moieties. Notably, in PFBT-COOH/PCBM and PFBT-T/PCBM binary Pdots where PCBM is used as a molecular electron acceptor, the fluorescence emissions of PFBT-COOH and PFBT-T are oxidatively quenched by 96% and 93% respectively, which is more efficient than the reductive quenching efficiency by TEMPO moieties. A similar observation can be seen in the fluorescence lifetime study of Pdots systems by employing a streak camera (Fig. 3c, d). The average lifetimes ($\tau_{ave}$) of PFBT-COOH Pdots

and PFBT-T Pdots are determined to be 285 and 240 ps, respectively (Supplementary Table 1). The shorter lifetime of PFBT-T Pdots as compared to that of PFBT-COOH Pdots is contributed by the hole transfer from PFBT* to TEMPO units. In binary Pdots, $\tau_{ave}$ of PFBT-COOH and PFBT-T are quenched significantly to 81 ps (PFBT-COOH/PCBM Pdots) and 41 ps (PFBT-T/PCBM Pdots). It is worthwhile to mention that the occurrence of Föster resonance energy transfer can be excluded by the negligible overlap between the fluorescence spectra of PFBT-COOH or PFBT-T and the absorption spectrum of PCBM (Fig. 3a), suggesting that the dominant fluorescence quenching contributed by PCBM is most likely from charge transfer. The charge transfer rate constants (based on $\tau_{ave}$, see details in Supplementary Table 1) between PFBT* and PCBM in PFBT-COOH/PCBM Pdots ($8.8 \times 10^9 s^{-1}$) and PFBT-T/PCBM Pdots ($2.0 \times 10^{10} s^{-1}$) are much larger compared to that between PFBT* and grafted TEMPO moieties ($6.7 \times 10^8 s^{-1}$) in PFBT-T Pdots, pointing out that oxidative quenching of PFBT* by PCBM should be the first step in binary Pdots systems, followed by reductive quenching with TEMPO, despite those TEMPO moieties are grafted in the polymer backbone.

In previous studies of Pdots systems, redshifts in fluorescence over time were observed. This is due to the energy disorder generated from molecule twisting[48]. This is important for fully understanding the origin of fluorescence quenching in PFBT-T because the fluorescence quenching in organic molecules can be affected by structure twisting[49]. To understand the effect of TEMPO linkage on the twisting of PFBT molecules, time-resolved fluorescence emission spectra of PFBT-COOH and PFBT-T dots were extracted and shown in Fig. 3e, f. The fluorescence emission peak of PFBT-COOH Pdots shifts from 542 to 552 nm after 520 ps, which is much smaller than that of PFBT Pdots. As for PFBT-T Pdots, a similar shift of 11 nm (from 529 to 540 nm) is observed, excluding the effect of TEMPO linkage on polymer twisting and further confirming that the quenching by TEMPO is contributed to charge transfer rather than polymer configuration changes.

The diffused TEMPO-NH₂ in THF and water can also quench PFBT* as shown in Fig. 4a, b and Supplementary Fig. 16. To compare the quenching ability of grafted TEMPO in PFBT-T polymer and diffused TEMPO-NH₂ in solutions, we have determined the amount of diffused TEMPO-NH₂ needed to reach the same quenching efficiency of the grafted TEMPO in PFBT polymer. As shown in Fig. 4a, to achieve 61% quenching of PFBT* fluorescence in PFBT-T by the linked TEMPO units, the concentration of TEMPO-NH₂ needed in THF is 7 mM and an obvious absorption peak of TEMPO-NH₂ from 250 to 300 nm is therefore observed in the solution (Supplementary Fig. 17). A similar phenomenon can be seen in PFBT-COOH Pdots. To quench ~ 47% of PFBT-COOH Pdots (-10 μg mL⁻¹) fluorescence, the TEMPO-NH₂ concentration needs to be higher than 3.5 mM in aqueous solution which also absorbs obviously from 250 to 300 nm (Supplementary Fig. 17). Notably, when calculating the required number of TEMPO-NH₂

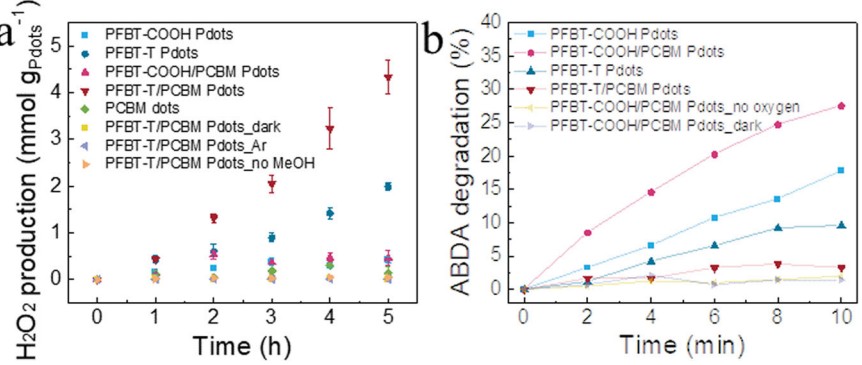

**Fig. 5 | Photocatalytic performance measurements. a** Photocatalytic performance of various Pdots systems, error bars show standard deviations between three samples. **b** Singlet oxygen generation experiment with ABDA as the probe.

molecules for quenching one mole of PFBT* to a certain degree in THF (61% quenching efficiency) and Pdots (47% quenching efficiency) (Fig. 4c), the number of covalently linked TEMPO needed shows over 1300-fold (in THF) and 400-fold (in Pdots) decrease as compared to the diffused TEMPO-NH$_2$ in solution. Therefore, the covalently linked TEMPO moieties can significantly facilitate the fluorescence quenching of PFBT* due to its short distance from PFBT backbone, indicating thousands and hundreds of times higher efficiency of hole transfer in this case compared to the diffused TEMPO-NH$_2$ molecules in THF and Pdots solutions.

Before studying photocatalytic H$_2$O$_2$ production, a chopped light experiment with PFBT-COOH and PFBT-T coated TiO$_2$ mesoporous film on FTO glass was conducted to verify the capability of photocatalytic MeOH oxidation of PFBT-T. TiO$_2$ is employed as the electron transfer layer due to its more positive conduction (−0.5 V vs. NHE) compared to the reduction potentials of PFBT-COOH and PFBT-T[50]. As from the CV results of the PFBT-T coated electrode (Supplementary Fig. 18), the oxidation peak of TEMPO disappears and the oxidation current increases after adding MeOH, illustrating that the oxidized TEMPO moieties on PFBT-T are capable of catalyzing MeOH oxidation in neutral condition. A similar observation is achieved by conducting the chopped light experiment (Supplementary Fig. 19), where the contribution of charge transfers from oxidized PFBT (PFBT$^{+*}$) to TEMPO and consequently to the MeOH oxidation process is strongly proved. This observation supports the charge transfer sequence illustrated above that PFBT* gives electrons to PCBM and the generated PFBT$^{+*}$ accepts electrons from TEMPO radicals can lead to the catalysis of MeOH oxidation.

To verify the contribution of the above process to the final photocatalytic H$_2$O$_2$ production and MeOH oxidation process, photocatalysis with Pdots is conducted in a water/MeOH (4/1, v/v) system at pH=7.4 (Fig. 5a). For PFBT-COOH, PFBT-COOH/PCBM Pdots and PCBM dots, negligible H$_2$O$_2$ was detected within 5 h photocatalytic reaction, this is consistent with our previous report that the MeOH oxidation remains sluggish at this pH[34]. When PFBT-T and PFBT-T/PCBM Pdots were utilized as photocatalysts, the systems achieved 378 and 865 μmol h$^{-1}$ g$^{-1}$$_{Pdots}$ H$_2$O$_2$ generation rate, respectively. PFBT-T/PCBM Pdots render an external quantum yield (EQE) of 0.75% at 450 nm. By conducting control experiments, light, O$_2$ and MeOH are all necessary for the photocatalytic generation of H$_2$O$_2$ to take place, meaning that the generation of H$_2$O$_2$ production with PFBT-T and PFBT-T/PCBM Pdots is from a photocatalytic oxygen reduction reaction where oxygen consumes photogenerated electrons to produce H$_2$O$_2$. When MeOH was replaced with other alcohols such as ethanol and benzyl alcohol in the PFBT-T/PCBM Pdots system, the system also showed photocatalytic H$_2$O$_2$ and aldehyde production activity (Supplementary Fig. 20), proving the universality of the TEMPO grafting strategy in polymer for

photocatalytic H$_2$O$_2$ product coupled with alcohol oxidation into corresponding aldehyde.

Unlike the system in our previous study[34], the oxidation of MeOH in the neutral condition is less likely to produce formic acid due to the restrained Cannizzaro reaction, and formaldehyde is the product instead of catalysis by TEMPO. As a result, formaldehyde was detected and the ratio of produced formaldehyde and H$_2$O$_2$ was determined to be 1:1, supporting that the H$_2$O$_2$ generation is from oxygen reduction which needs two electrons eventually involved (Supplementary Fig. 21). By adding benzoquinone (BQ) as the scavenger of superoxide radicals (O$_2^{·-}$)[51], the H$_2$O$_2$ generation with PFBT-T/PCBM Pdots was completely inhibited while the formaldehyde production remained (Supplementary Table 2), further demonstrating that the photocatalytic H$_2$O$_2$ should go through a two-step one-electron transfer pathway with O$_2^{·-}$ as an intermediate.

It is important to point out that the residual Pd in polymer synthesized by Suzuki-coupling reactions has been proposed to play a significant role in photocatalytic H$_2$ production[52], posing the problem that residual Pd in PFBT-T could play a role in O$_2$ reduction during photocatalytic H$_2$O$_2$ production as well. However, recently we have reported that in the presence of PCBM as the electron acceptor, the electron received by PCBM from PFBT* will not be able to directly transfer to Pd if Pd is the catalyst for proton reduction[53]. Moreover, the residual Pd was proved to be negligible in contributing to photocatalytic H$_2$O$_2$ production in alkaline conditions which also involves the O$_2$ reduction step[34]. Therefore, based on all these findings, the residual Pd in PFBT-T (388 ppm) should not play an important role in H$_2$O$_2$ formation.

In our previous study, the stability of PFBT/PCBM Pdots system in photocatalytic oxygen reduction is a remaining challenge to be addressed[34]. The generation of singlet oxygen is considered to be the main reason for the degradation of organic molecules[34]. Surprisingly, grafting TEMPO in the polymer dramatically promotes the stability of PFBT-T in photocatalysis as compared to PFBT-COOH systems (Supplementary Fig. 22, 23). To understand the origin of the increased stability, 9,10-Anthracenediyl-bis(methylene)dimalonic acid (ABDA) is utilized as the probe of singlet oxygen (Supplementary Fig. 24). As shown in Fig. 5d, with the removal of oxygen and maintaining a dark environment, ABDA degradation in PFBT-COOH/PCBM system is nearly fully suppressed, proving that the light-induced singlet oxygen generation is the main origin of ABDA degradation. The degradation of ABDA in PFBT-COOH and PFBT-COOH/PCBM Pdots systems is faster than that in PFBT-T and PFBT-T/PCBM systems in presence of oxygen, indicating that the generation of singlet oxygen in the later systems is effectively suppressed (see detailed mechanism in SI). Therefore, the stability of PFBT-T and PFBT-T/PCBM Pdots during photocatalysis is promoted by suppressing singlet oxygen generation due to efficient charge extraction in the system.

## Discussion

Organic molecular catalyst TEMPO has been grafted to PFBT skeleton in the polymer PFBT-T which has been used to prepare binary polymer dots (Pdots) in the presence of PCBM as a molecular electron acceptor. Owing to the efficient intramolecular charge transfer between PFBT backbone and grafted TEMPO in PFBT-T and efficient intermolecular charge transfer between PFBT backbone and PCBM, the binary PFBT-T/PCBM Pdots system has shown photocatalytic $H_2O_2$ production rate and a stoichiometric formaldehyde generation rate of 865 μmol $h^{-1}$ $g^{-1}_{Pdots}$ in neutral pH condition, with an EQE of 0.75% at 450 nm. Contributed by TEMPO grafting, the photocatalytic stabilities of PFBT-T and PFBT-T/PCBM Pdots have been dramatically improved. By replacing MeOH with other alcohols, the system still works well to produce $H_2O_2$ coupled with production of corresponding aldehydes, showing a universal application of the system in alcohol oxidation and $H_2O_2$ production. This work proves an idea of grafting efficient molecular catalysts to polymer structures and fabricating Pdots systems to accelerate sluggish chemical reactions, which provides Pdots systems with opportunities to be utilized in more practical applications such as chemical synthesis and fuel production together with molecular catalysts.

## Methods

### Synthesis of monomer (2,2'-(2,7-dibromo-9H-fluorene-9,9-diyl) diacetic acid)

The synthesis of the monomer is based on a reported method with simple modifications. Ethyl bromoacetate (15 g, 89.8 mmol) was diluted with DMSO (30 ml) and added dropwise to a solution of 9.6 g (29.6 mmol) of 2,7-dibromofluorene and sodium hydroxide (50% w/w) aqueous solution (10 mL) in DMSO (250 mL) under nitrogen at 273 K. After the addition, the resulting solution was stirred for 12 h at room temperature. Afterward, the solid product was collected by filtration and dissolved in 100 mL water. Then 10 M HCl (10 mL) was added dropwise to the reaction mixture in ice-water bath. The resulting solution was stirred for 30 min. Precipitate was collected by filtration, followed by washing with water for three times, then the crude product was dried in vacuum at 323 K. Finally, the white bulk product (3.1 g, 6.9 mmol, 23.3% yield) was collected and used for next steps.

### Synthesis of PFBT

A mixture solution of 2,7-Dibromofluorene (330 mg, 1.0 mmol), 2,7-Dibromo-9,9-di-n-octylfluorene (548 mg, 1.0 mmol) and 4,7-Bis(4,4,5,5-tetramethyl-1,3,2-dioxaborolan-2-yl)-2,1,3-benzothiadiazole (776 mg, 2.0 mmol) in 100 mL DMF was first degassed with Ar for 30 min. Afterwards, Pd(PPh₃)₄ (10 mg) and K₂CO₃ (2 M, 2 mL) were added consecutively, followed by degassing for another 10 min. The reaction mixture was stirred and heated at 393 K for 24 h under Ar atmosphere. After the reaction, the mixture was washed with 100 mL CH₂Cl₂ for 3 times and the collected CH₂Cl₂ solution was washed with 150 mL water for 3 times. The combined organic layer was dried under vacuum. The collected bulk product was washed with methanol and acetone 3 times to get 415 mg (0.19 mmol) of product (Mn=2.2k), the yield is 4.8%.

### Synthesis of PFBT-COOH

For the synthesis of PFBT-COOH. A mixture solution of monomer (432 mg, 1.0 mmol), 4,7-Bis(4,4,5,5-tetramethyl-1,3,2-dioxaborolan-2-yl)-2,1,3-benzothiadiazole (760 mg, 2.0 mmol) and 2,7-Dibromo-9,9-di-n-octylfluorene (542 mg, 1.0 mmol) in DMF (100 mL) was first degassed with Ar for 30 min. Afterwards, Pd(PPh₃)₄ (10 mg) and K₂CO₃ (2 M, 2 mL) were added consecutively, followed by degassing for another 10 min. The reaction mixture was stirred and heated at 393 K for 24 h under Ar atmosphere. After the reaction, the mixture was washed with 100 mL CH₂Cl₂ 3 times and the collected CH₂Cl₂ solution was washed with 150 mL water 3 times. The combined organic layer was dried under a vacuum. The collected bulk product was washed with methanol and acetone 3 times to get 190 mg (0.08 mmol) of product (Mn=2.3k), the yield is 2.1%.

### Synthesis of PFBT-T

To link TEMPO with the polymer, a solution of PFBT-COOH (150 mg, 0.065 mmol) in 100 mL SOCl₂ was stirred and refluxed for 48 h. Afterwards, SOCl₂ was removed by distillation. Then 20 mg 4-Amino-2,2,6,6-tetramethylpiperidine-1-oxyl (TEMPO-NH₂) (0.12 mmol) in 30 mL dry THF was added and 1 mL triethylamine was added dropwise and slowly. The reaction mixture was stirred for 24 h at room temperature. THF was removed by vacuum evaporation and the product was washed with 100 mL water 10 times and reprecipitated in ethyl easter twice to remove the unreacted TEMPO-NH₂. The product (37 mg, 0.016 mmol) was collected after vacuum drying overnight, the yield is 24.7%.

### Pdots preparation

Taking PFBT Pdots as an example, PFBT-T and polystyrene grafted with ethylene oxide and carboxyl groups (PS-PEG-COOH) were separately dissolved in THF to make stock solutions in concentrations of 0.05 mg mL⁻¹ and 1 mg mL⁻¹ respectively. Then, 10 mL PFTB-T solution and 1.5 mL PS-PEG-COOH were mixed. The mixture was poured into 25 mL of deionized water and then the formed dark yellow solution was heated with a water bath at 80 °C for fast evaporation of THF. After complete evaporation of THF, an aqueous Pdots solution was obtained by filtering the solution with 0.45 μm pore size PVDF syringe filters.

### Photocatalysis experiments

Photocatalytic $H_2O_2$ generation reaction proceeded in a quartz cuvette. In detail, 2.5 mL Pdots reaction solution (22 μg mL⁻¹) which contains 5 M MeOH was purged by O₂ for 15 min to ensure the saturation of O₂ in solution (pH=7.4). Afterwards, the cuvette was irradiated by a LED lamp (Zenaro Lighting GmbH, SL-PAR38B/P17/50/E50/ND/27/UNI/EU/ZN, 420–750 nm) with an irradiation density of 50 mW cm⁻². The irradiation density was measured with a PM100D power meter (Thorlabs).

## Data availability

All data supporting the findings of this study are available within the article and the Supplementary Information. Raw data are provided with this paper. Source data are provided with this paper.

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

## Acknowledgements

We gratefully thank the K&A Wallenberg Foundation, Academy Fellow program for financial support (2019.0156). We would like to thank Prof. Tim Melander Bowden for offering help in multiple GPC measurements and Dr. Vitalii Shtender for offering help in ICP-OES measurement. Also, we would like to thank Dr. Martin Axelsson, Catherine Ellen Johnson, Andjela Brnovic and Dr. Hongwei Song for their helpful discussion. We would like to acknowledge the use of the Cryo-EM Uppsala facility for cryogenic electron microscopy measurements.

## Author contributions

S.W. conducted all synthesis, catalysis and partial spectroscopy studies and wrote the manuscript. M.P. conducted the Cryo-EM measurement. X.Z. conducted the streak camera measurement. P.H. conducted the EPR measurement. B.C. conducted the NMR measurement and helped during molecular synthesis. O.S. conducted the GPC measurement. H.T. supervised and directed the project. All authors contributed to data analysis and commented on the manuscript.

## Funding

## Competing interests

The authors declare no competing interests.
