## [Peer Review File · Nature Communications]

Covalently linked molecular catalysts in conjugated polymer dots boost photocatalytic alcohol oxidation in neutral conditionREVIEWER COMMENTS

Reviewer #1 (Remarks to the Author):

In the present work, Wang et al. report an organic donor-acceptor photocatalyst (pdot) with an embedded molecular catalyst which produces H₂O₂ while performing MeOH oxidation at neutral pH. While the neutral pH operation makes this work appealing, there are several issues and queries which I would like to see addressed prior to publication:

1) The authors calculate an excited state reduction potential for *PFBT-T**/*PFBT-T*– by summing the ground state reduction potential and optical transition energy. Generally, excited state redox potentials are not simply linear combinations of these properties; why do the authors believe that this approach is valid here?

2) In their previous work on PFBT/PCBM pdots (<https://doi.org/10.1002/anie.202202733>), the authors report EQEs of 30% (5 min) and 14% (75min) at 450nm. Why are EQEs for the current TEMPO system so much lower (0.75% at 450 nm)? Is this due to the neutral pH? If so, what pathways exist towards higher EQEs at neutral pH?

3) It is unclear where the values in the energy alignment diagram (Figure 3b) come from. Please clarify and add details to manuscript or SI.

4) Residual palladium has been found to act as the primary catalyst for hydrogen evolution in fluorene-based polymer photocatalysts when prepared via Pd-catalyzed cross coupling reactions (<http://doi.wiley.com/10.1002/aenm.201802181>). Did the authors quantify residual metal content in their pdots, for example via ICP-MS? The higher electrocatalytic activity of PFBT-T seems to point to catalytic activity of the TEMPO unit, but it should be ruled out that changes in the concentration of residual metals are responsible for the observed performance trends.

5) Covalent attachment of TEMPO is currently not supported conclusively and needs to be demonstrated more unambiguously.

- 5a) The authors report “After the linkage of TEMPO, C-N stretching from 1000 to 1200 cm⁻¹ was detected, indicating the successful grafting of TEMPO moieties in PFBT-T”; how do they distinguish this frequency from the C-N stretching within TEMPO?

- 5b) Based on the TEMPO solution quenching experiments it is clear that TEMPO residues are not simply floating in solution, but neither the EPR, CV, or UV/vis data seems to exclude the possibility of adsorbed TEMPO (rather than covalently attached). Can the authors exclude such non-covalent attachment? A convincing demonstration of covalent attachment would be to perform a PFBT-T synthesis without COOH functionalization, eliminating the possibility of covalent attachment of TEMPO. Any embedded TEMPO would be adsorbed, and the resulting material should exhibit very different properties (for example, in terms of fluorescence quenching and charge transfer efficiencies) from PFBT-T if the latter was indeed a covalent material.

6) The authors report “average lifetimes” for their transient PL experiments, but it is unclear how these are calculated. Do these values come from a weighted average of a multiexponential fit? If so, please add a table with the individual lifetime components to the SI.

7) Quenching efficiencies (QE; referred to as charge transfer quantum yields in the manuscript) are commonly calculated via (a) rate constants of excitonic decays with $QE = (k-k_0)/k$ where k_0 is the exciton decay rate constant in the absence of a quencher and k is the corresponding value in the presence of a quencher; or (b) using I_q/I_0 , where I_0 is the integrated PL intensity (time resolved or steady state) in the absence of a quencher and I_q is the corresponding value in the presence of a quencher. (a) and (b) should ideally give similar results. (a) seems to be the path which the authors are pursuing, but this approach is not valid if the used rate constants are weighted averages of multiple decay components. I suggest the authors focus only on k of one decay component, or better, use approach (b) instead to calculate quenching efficiencies. If the authors continue using rate constants, they should demonstrate that their values are consistent with those obtained via approach (b).

8) Do the authors have any hypotheses why TEMPO suppress singlet oxygen formation?

9) Please thoroughly check the correctness of references in the SI. For example, SI line 115 references the wrong figure.

Reviewer #2 (Remarks to the Author):

In this manuscript, the authors reported a binary PFBT-T/PCBM polymer dots system for photocatalytic hydrogen peroxide production and methanol oxidation in a neutral condition. Although grafting molecular catalysts onto polymer skeleton has seldom been studied in polymer dots systems, but as the authors said, it has been widely used in bulk organic photocatalyst systems, which makes the current work less meaningful. Moreover, the photocatalytic performance is unsatisfactory and the mechanism lacks in-depth analysis. Therefore, it is not suitable for publication in the top journal Nature Communications, and the specific comments are as follows:

1. The photocatalytic performance of the PFBT-T/PCBM polymer dots system is much lower compared with other organic photocatalysts such as graphitic carbon nitride and covalent organic frameworks in the same condition, even lower than their performance in pure water. Moreover, the separation of H₂O₂, methanol, formaldehyde, and other products is too complicated and requires ultrahigh energy and economic consumption. Therefore, it is hard to say if the PFBT-T/PCBM polymer dots system has promising practical applications.
2. The authors provide evidence for the preparation of PFBT-T and its intrinsic properties such as the enhanced stability and the boosted intramolecular charge transfer. However, the concrete mechanism that enables PFBT-T/PCBM photocatalytic activity in the neutral condition compared with PFBT/PCBM which shows activity only in alkaline conditions is still unclear and unconvincing.
3. In lines 260-262, they claimed “the oxidation of MeOH in neutral condition is less likely to

produce formic acid since the restrained Cannizzaro reaction, and formaldehyde is considered as the product instead.” It can be seen in Figure S3 that the “Adj. R-Square” of the calibration curve is only 0.9868, so it’s not rigorous to conclude that formaldehyde is the single oxidation product without any detection of formic acid. I do not think the selectivity of methanol oxidation can reach 100%.

4. In lines 264-265, they claimed “supporting that the H₂O₂ generation goes through a two-electron reduction of oxygen ($O_2 + 2e^- + 2H^+ \rightarrow H_2O_2$).” But, there is a lack of evidence for confirming whether the H₂O₂ generation goes through a one-step two-electron or two-step one-electron reduction of oxygen.

5. For the measurement of external quantum efficiency, the concentration of Pdots and the irradiation density become inconsistent with the photocatalytic activity test. What’s the purpose of changing these factors and why is 2.7 mW/cm² determined as the irradiation density?

6. The pH value change before and after the photocatalytic reaction is important, but not considered.

7. Many spelling mistakes must be carefully checked and corrected throughout the manuscript, such as “graphitic” in line 38 and “framework” in line 39.

Reviewer #3 (Remarks to the Author):

This is an interesting paper, but mostly because of the tethered TEMPO catalyst they have developed. This may be interesting in catalysis beyond the specific H₂O₂ generation that is the subject of this communication. In the context of the title “Photocatalytic Hydrogen Peroxide production ...” I am not convinced this is practical for the production of hydrogen peroxide, particularly with EQE of 0.75% it does not qualify as “outstanding production rates”, as suggested in page 4. At the end of this inefficient process one has a mixture of H₂O₂, methanol, formaldehyde and the catalyst-polymer. While the widely used anthraquinone method has its own separation challenges, is there an improvement here?

The materials used have been well characterized, as has the progress of the reaction. In contrast a key reagent, light, is poorly characterized, as for the two light sources used (LED lamp and Xenon lamp with filter) no spectrum is shown. Further, the irradiation density (irradiance) is just a number given with no indication on how it was determined. This would make this work hard to reproduce. If this paper is resubmitted, here or elsewhere, this issue must be resolved.

Response Letter

Dear reviewers,

Thank you very much for reviewing our manuscript of "*Covalently linked Organic Molecular Catalyst in Conjugated Organic Polymer Dots Boosts Photocatalytic Hydrogen Peroxide production and Methanol oxidation in Neutral Condition*" (NCOMMS-23-58020). All comments received are very helpful for us to improve the manuscript.

The manuscript now has been carefully revised according to all comments. The revised manuscript and supporting information with all changes highlighted in yellow are uploaded for review. Below we respond to the comments in detail.

Sincerely,

Haining Tian, on behalf of all co-authors

Reviewer #1 (Remarks to the Author):

In the present work, Wang et al. report an organic donor-acceptor photocatalyst (pdot) with an embedded molecular catalyst which produces H₂O₂ while performing MeOH oxidation at neutral pH. While the neutral pH operation makes this work appealing, there are several issues and queries which I would like to see addressed before publication:

1) The authors calculate an excited state reduction potential for *PFBT-T**/*PFBT-T*⁻ by summing the ground state reduction potential and optical transition energy. Generally, excited state redox potentials are not simply linear combinations of these properties; why do the authors believe that this approach is valid here?

Response by the author: Thanks for the question. The method used is widely used to estimate the reduction/oxidation potential of excited states of photosensitizers (Chem. Soc. Rev., 2022, 51, 6909–6935; ACS Appl. Mater. Interfaces 2018, 10, 13, 10828–10834; J. Am. Chem. Soc. 2022, 144, 30, 13600–13611; Org. Chem. 2007, 72, 25, 9550–9556; Phys. Chem. C 2010, 114, 10, 4738–4748; J. Mater. Chem. A, 2014, 2, 9944). Therefore, we used this method to analyze the feasibility of charge transfer in the PFBT-T system.

2) In their previous work on PFBT/PCBM pdots (<https://doi.org/10.1002/anie.202202733>), the authors report EQEs of 30% (5 min) and 14% (75min) at 450nm. Why are EQEs for the current TEMPO system so much lower (0.75% at 450 nm)? Is this due to the neutral pH? If so, what pathways exist towards higher EQEs at neutral pH?

Response by the author: This is a very good question! There are two main reasons for the drop in photocatalytic activity in neutral pH. Firstly, we have demonstrated in this work and our

previous work that the oxidation of MeOH by the oxidized PFBT is not feasible in neutral pH. Therefore, in this work, considering that the concentration of TEMPO grafting is low (14 % w/w) and only the oxidized TEMPO molecules can contribute to oxidizing MeOH, the overall photocatalytic reaction is therefore slow. In previous work, MeOH can be deprotonated in alkaline conditions to become more reductant and therefore was oxidized efficiently. Secondly, as the hydrophilic chains in surfactant used in Pdots locate outside of Pdots cores and hydrophobic PFBT-T and PCBM molecules sit in the core region, the interaction of oxidized TEMPO species with MeOH is very important. Hence, the low activity of the PFBT-T system can also be explained by the unsatisfactory penetration of MeOH into Pdots. Although the efficiency is lower as compared to the previous work, the system working in neutral conditions is an important step.

To further improve the efficiency, in our future work we will work on putting TEMPO units in a position where they can efficiently accept holes from the polymer and at the same time MeOH can well interact with them. The strategies are to optimize the morphology of Pdots using different surfactants or to the catalysts to the hydrophilic side of surfactants which can locate the TEMPO units near the surface of Pdots.

3) It is unclear where the values in the energy alignment diagram (Figure 3b) come from. Please clarify and add details to the manuscript or SI.

Response by the author: We have added a detailed description in SI, also shown below:

Estimation of band gaps and energy level calculation of polymers

The optical bandgap E_g of polymers, was estimated by zero-zero transition energy (E_{0-0}), which is calculated by $E_{0-0} = hc/\lambda = 1240/\lambda$. Where h is Planck's constant, c is the speed of light and λ is the wavelength.

λ , ca. 497 nm for PFBT-T, is determined as the intersection point of the UV-vis spectra and the steady state emission spectra of PFBT-T in THF. Therefore, the $E_g = E_{0-0} = 1240/\lambda = 2.49$ eV.

The reduction potential ($E_{PFBT-T/PFBT-T^-}$) of PFBT-T was measured to be -0.95 V vs. NHE in THF solution with CV measurements (Figure S10). Reduction potential of the excited PFBT-T ($E_{PFBT-T^*/PFBT-T^-}$) was calculated based on equation below:

$$E_{PFBT-T^*/PFBT-T^-} = E_{PFBT-T/PFBT-T^-} + E_{0-0} \quad (1)$$

Therefore, ($E_{PFBT-T^*/PFBT-T^-}$) was calculated to be 1.54 V vs.NHE.

For energy levels of PCBM, TEMPO oxidation and O₂ reduction, relative references have been cited in the revised manuscript.

4) Residual palladium has been found to act as the primary catalyst for hydrogen evolution in fluorene-based polymer photocatalysts when prepared via Pd-catalyzed cross-coupling reactions (<http://doi.wiley.com/10.1002/aenm.201802181>). Did the authors quantify residual metal content in their pdots, for example via ICP-MS? The higher electrocatalytic activity of PFBT-T seems to point to catalytic activity of the TEMPO unit, but it should be ruled out that changes in the concentration of residual metals are responsible for the observed performance trends.

Response by the author: Thank you for this comment! This is a very common concern for researchers who use polymer materials synthesized from the Suzuki Coupling reaction, to do catalysis without the addition of a co-catalyst. According to your suggestion, we have quantified the content of Pd in PFBT-COOH and PFBT-T and the results are added in the revised manuscript. Based on our recent work (*JACS Au* 2024, <https://doi.org/10.1021/jacsau.3c00681>), in the presence of PCBM, the electron received by PCBM from PFBT will not be able to transfer to Pd if Pd is the catalyst for proton reduction. The statement is also valid in the current work for O₂ reduction. Similarly, this is the reason we observed no change in photocatalytic activity of H₂O₂ production with PFBT-PCBM Pdots consisting of unwashed PFBT and washed PFBT (Pd < 10ppm) in our previous work (*Angew. Chem.Int. Ed.* 2022, 61, e202202733).

The paragraph shown below has been added to the revised manuscript.

‘It is necessary to point out that residual Pd in polymer synthesized by Suzuki-coupling reaction could play a significant role in photocatalytic H₂ production, (*ACS Energy Lett.* 2018, 3(11), 2846–2850) posing the problem that residual Pd in PFBT-T could play a role in O₂ reduction during photocatalytic H₂O₂ production. However, recently we have reported that in the presence of PCBM as the electron acceptor, the electron received by PCBM from PFBT* will not be able to transfer to Pd if Pd is the catalyst for proton reduction. (*JACS Au*, 2024, 4(2), 570–577) Moreover, the Pd residual was proved to be negligible in contributing to photocatalytic H₂O₂ production in alkaline conditions which also involves the O₂ reduction step.³³ Therefore, based on all these findings, the residual Pd in PFBT-T (388 ppm) does not play an important role in H₂O₂ formation.’

5) Covalent attachment of TEMPO is currently not supported conclusively and needs to be demonstrated more unambiguously.

• 5a) The authors report “After the linkage of TEMPO, C-N stretching from 1000 to 1200 cm⁻¹ was detected, indicating the successful grafting of TEMPO moieties in PFBT-T”; how do they distinguish this frequency from the C-N stretching within TEMPO?

• 5b) Based on the TEMPO solution quenching experiments it is clear that TEMPO residues are not simply floating in solution, but neither the EPR, CV, or UV/vis data seems to exclude the possibility of adsorbed TEMPO (rather than covalently attached). Can the authors exclude such non-covalent attachment? A convincing demonstration of covalent attachment would be to perform a PFBT-T synthesis without COOH functionalization, eliminating the possibility of covalent attachment of TEMPO. Any embedded TEMPO would be adsorbed, and the resulting material should exhibit very different properties (for example, in terms of fluorescence quenching and charge transfer efficiencies) from PFBT-T if the latter was indeed a covalent material.

Response by the author: Thank you for the comment. The linked TEMPO in TEMPO has been well characterized from IR, EPR and electrochemistry. To make the characterization clearer, regarding comment 5a, we have revised the description in the manuscript as ‘After the linkage of TEMPO, C-N stretching from 1000 to 1200 cm^{-1} was detected, indicating the **existence** of TEMPO moieties in PFBT-T.’ since the covalent grafting of TEMPO cannot be proved via this stretching signal. As for comment 5b, we have synthesized the PFBT polymer which has a very similar molecule weight as PFBT-COOH. Also, we have done the synthesis which is possible to result in the adsorption of TEMPO molecules on polymers. The synthesis process has been added in the revised SI:

‘Synthesis of PFBT

A mixture solution of 2,7-Dibromofluorene (330 mg, 1 mmol), 2,7-Dibromo-9,9-di-n-octylfluorene (548 mg, 1 mmol) and 4,7-Bis(4,4,5,5-tetramethyl-1,3,2-dioxaborolan-2-yl)-2,1,3-benzothiadiazole (776 mg, 2 mmol) in 100 mL DMF was first degassed with Ar for 30 min. Afterwards, $\text{Pd}(\text{PPh}_3)_4$ (10 mg) and K_2CO_3 (2 M, 2 mL) were added consecutively, followed by degassing for another 10 min. The reaction mixture was stirred and heated at 393 K for 24 h under Ar atmosphere. After the reaction, the mixture was washed with 100 mL CH_2Cl_2 for 3 times and the collected CH_2Cl_2 solution was washed with 150 mL water for 3 times. The combined organic layer was dried under a vacuum. The collected bulk product was washed with methanol and acetone 3 times to get 415 mg of product ($M_n=2.2k$).

Synthesis of PFBT with possibly adsorbed TEMPO

In this process, 50 mg PFBT and 10 mg TEMPO-NH₂ were mixed in 50 mL THF and stirred at room temperature for 24 h. THF was removed by vacuum evaporation and the product was washed with water under sonication for 30 min in 250 mL water three times to remove the water-soluble TEMPO-NH₂ residuals on the surface. Afterwards, the washed PFBT was dissolved in THF and precipitated in ethyl ether twice to remove the capsulated TEMPO-NH₂.

After the synthesis, we also checked the absorption spectra and fluorescence quenching results of the samples. The adsorption of TEMPO-NH₂ molecules can be excluded because we didn’t see any shift in absorption spectra or any fluorescence quenching. The paragraph below has been added in the revised SI:

Figure S6. Absorption (a) and fluorescent emission (b) of PFBT with possibly adsorbed TEMPO-NH₂.

‘As shown in Figure S6a, no shift in absorption has been observed after synthesis, excluding the possibility of covalent grafting of TEMPO in this situation. According to the steady-state quenching results in Figure S6b, the adsorption of TEMPO-NH₂ molecules on PFBT is excluded since no quenching on fluorescence intensity was observed.’

We have also added a short description in the revised manuscript:

‘The possibility of TEMPO-NH₂ molecule physical adsorption is excluded by conducting a control synthesis and relevant spectroscopic studies (see details in Figure S6).’

6) The authors report “average lifetimes” for their transient PL experiments, but it is unclear how these are calculated. Do these values come from a weighted average of a multiexponential fit? If so, please add a table with the individual lifetime components to the SI.

Response by the author: We have added the table shown below to the revised SI. The average lifetime comes from a weighted average of a biexponential fit.

Table S1. Lifetime parameters of PFBT-COOH, PFBT-T, PFBT-COOH/PCBM and PFBT-T/PCBM Pdots.

Sample	τ_1 (ps)	Rel ₁ (%)	τ_2 (ps)	Rel ₂ (%)	τ_{ave} (ps)
PFBT-COOH	80	39	415	61	285
PFBT-T	73	44	370	56	240
PFBT-COOH/PCBM	31	81	295	19	81
PFBT-T/PCBM	23	90	198	10	41

7) Quenching efficiencies (QE; referred to as charge transfer quantum yields in the manuscript) are commonly calculated via (a) rate constants of excitonic decays with $QE = (k-k_0)/k$ where k_0 is the exciton decay rate constant in the absence of a quencher and k is the corresponding value in the presence of a quencher; or (b) using I_q/I_0 , where I_0 is the integrated PL intensity (time-resolved or steady state) in the absence of a quencher and I_q is the corresponding value in the

presence of a quencher. (a) and (b) should ideally give similar results. (a) seems to be the path which the authors are pursuing, but this approach is not valid if the used rate constants are weighted averages of multiple decay components. I suggest the authors focus only on k of one decay component, or better, use approach (b) instead to calculate quenching efficiencies. If the authors continue using rate constants, they should demonstrate that their values are consistent with those obtained via approach (b).

Response by the author: Thanks for this comment. We have deleted the quenching efficiency data which was calculated based on averaged fluorescence lifetimes in the revised manuscript. Now the quenching efficiencies are calculated only based on steady-state fluorescence data as shown below:

‘As compared to PFBT-COOH, the fluorescence intensity is reduced by 61 % in PFBT-T (Figure S11a). Moreover, the averaged lifetime of fluorescence is quenched from 2.2 ns in PFBT-COOH to 1.6 ns in PFBT-T (Figure S11b). This phenomenon is also observed in Pdots phases where the fluorescent emission of PFBT-T Pdots is quenched by 47 % after TEMPO grafting (Figure 3a) as compared to PFBT-COOH Pdots, proving that the PFBT* in PFBT-T can be reductively quenched by the grafted TEMPO moieties. Notably, in PFBT-COOH/PCBM and PFBT-T/PCBM binary Pdots where PCBM is used as a molecular electron acceptor, the fluorescence emissions of PFBT-COOH and PFBT-T are oxidatively quenched by 96 % and 93 % respectively, which is more efficient than the reductive quenching efficiency by TEMPO moieties.’

8) Do the authors have any hypotheses as to why TEMPO suppress singlet oxygen formation?

Response by the author: Thanks for this question. In our previous work, we found that the singlet oxygen formation rate increased as the pH decreased. Moreover, the formation of singlet oxygen can be significantly hindered in the presence of MeOH. Based on this, we concluded that the MeOH can consume the holes from Pdots and then suppress the energy transfer process from PFBT to oxygen forming singlet oxygen. In this work, we have grafted TEMPO in polymer and TEMPO can efficiently consume holes from excited or reduced polymer and then used to oxidize MeOH, therefore suppressing the formation of singlet oxygen. The description below has been added in the revised SI.

‘As discussed in our previous work,⁴ the formation of singlet oxygen is competing with oxidation of MeOH because consumption of the generated holes in Pdots by MeOH will suppress the Dexter energy transfer pathway to O₂ forming singlet oxygen. (*J. Phys. Chem. C* 2018, 122, 14, 7824–7830) In PFBT-T and PFBT-T/PCBM systems, the generation rate of singlet oxygen is slower than PFBT and PFBT/PCBM systems, which is contributed by efficient MeOH oxidation from the TEMPO in PFBT-T and PFBT-T/PCBM.’

9) Please thoroughly check the correctness of references in the SI. For example, SI line 115 references the wrong figure.

Response by the author: The SI has been carefully checked and all relative references have been corrected.

Reviewer #2 (Remarks to the Author):

In this manuscript, the authors reported a binary PFBT-T/PCBM polymer dots system for photocatalytic hydrogen peroxide production and methanol oxidation in a neutral condition. Although grafting molecular catalysts onto polymer skeleton has seldom been studied in polymer dots systems, but as the authors said, it has been widely used in bulk organic photocatalyst systems, which makes the current work less meaningful. Moreover, the photocatalytic performance is unsatisfactory and the mechanism lacks in-depth analysis. Therefore, it is not suitable for publication in the top journal Nature Communications, and the specific comments are as follows:

1. The photocatalytic performance of the PFBT-T/PCBM polymer dots system is much lower compared with other organic photocatalysts such as graphitic carbon nitride and covalent organic frameworks in the same condition, even lower than their performance in pure water. Moreover, the separation of H₂O₂, methanol, formaldehyde, and other products is too complicated and requires ultrahigh energy and economic consumption. Therefore, it is hard to say if the PFBT-T/PCBM polymer dots system has promising practical applications.

Response by the author: Thank you for this comment. The main achievement of this work is not to have tremendous improvements in photocatalytic activity, but rather to demonstrate a novel design of polymer linked with a molecular catalyst for alcohol oxidation coupled with hydrogen peroxide production. Therefore, the design in this work is ground-breaking in Pdots systems and even in polymeric photocatalysis systems. This design strategy can also be used in systems which already have high photocatalytic activities because the oxidation of organic molecules (e.g. alcohols) is always the rate-limiting step. The further improvement of the system in this work can be achieved by optimization work such as increasing the TEMPO grafting concentration and morphology controlling which we are planning in our future work. We hope that the reviewer can understand not all research at an early stage always shows a strong application perspective, a novel solution to a scientific question should be appreciated.

2. The authors provide evidence for the preparation of PFBT-T and its intrinsic properties such as the enhanced stability and the boosted intramolecular charge transfer. However, the concrete mechanism that enables PFBT-T/PCBM photocatalytic activity in the neutral condition compared with PFBT/PCBM which shows activity only in alkaline conditions is still unclear and unconvincing.

Response by the author: Thanks for the comment. The reason why PFBT/PCBM Pdots only show activity in alkaline conditions has been illustrated in our previous work (*Angew. Chem. Int. Ed.* 2022, 61, e202202733). For PFBT-T/PCBM Pdots, we have shown in the manuscript that the hole transfer from PFBT* to TEMPO radicals is efficient in quenching 61 % and 47 % of the fluorescence of PFBT* in THF and Pdots respectively. The oxidized TEMPO moieties are responsible for MeOH oxidation activity under neutral conditions. In order to give a detailed mechanism, we have added the reaction mechanism scheme in the revised SI.

Figure S16. Reaction mechanism of photocatalytic H_2O_2 and formaldehyde production PFBT-T/PCBM Pdots in neutral conditions.

3. In lines 260-262, they claimed “the oxidation of MeOH in the neutral condition is less likely to produce formic acid since the restrained Cannizzaro reaction, and formaldehyde is considered as the product instead.” It can be seen in Figure S3 that the “Adj. R-Square” of the calibration curve is only 0.9868, so it’s not rigorous to conclude that formaldehyde is the single oxidation product without any detection of formic acid. I do not think the selectivity of methanol oxidation can reach 100%.

Response by the author: Thank you very much for this comment. We have done a more completed calibration curve for detection of HCHO and it has been added in the revised SI.

Figure S5. Calibration curve of detection of formaldehyde.

In the repeated calibration curve, the Adj. R-square has reached 0.99886 and the calculated generation of HCHO is also very close to the original value. TEMPO is very well-known to selectively oxidize alcohols to aldehydes under neutral conditions which is widely reported

(*Eur. J. Org. Chem.* 2020, 2399–2408; *Green Chem.*, 2011, 13, 905; *Acc. Chem. Res.* 2002, 35, 9, 774–781). Based on the fact that PFBT cannot oxidize MeOH under neutral conditions, the only oxidation pathway of MeOH is through oxidized TEMPO which can only generate formaldehyde as the product. Moreover, the disproportionation reaction of formaldehyde (Cannizzaro reaction) can only happen in the presence of a strong base (e.g. NaOH and KOH) which is lacking in this work. Therefore, formaldehyde is the only product in the PFBT-T/PCBM system.

4. In lines 264-265, they claimed “supporting that the H₂O₂ generation goes through a two-electron reduction of oxygen (O₂+2e⁻+2H⁺→H₂O₂).” However, there is a lack of evidence for confirming whether the H₂O₂ generation goes through a one-step two-electron or two-step one-electron reduction of oxygen.

Response by the author: Thanks for the comment. This comment refers to whether the electron can accumulate in Pdots because one-step two-electron reduction of oxygen requires the same PCBM molecule to accumulate two electrons and transfer these two electrons to the same adsorbed oxygen molecule on this specific PCBM molecule. However, in molecular catalysts, the recombination of the electron from the first excitation and the hole generated during the second excitation can be very fast (*Energy Fuels* 2021, 35, 23, 18848–18856). In other words, electrons are less likely to accumulate in molecule catalysts. Therefore, the H₂O₂ generation should go through a two-step one-electron reduction of oxygen, which is also illustrated in many other works (*Nature Catalysis* volume 4, pages374–384 (2021); *ACS Catal.* 2020, 10, 6, 3697–3706).

5. For the measurement of external quantum efficiency, the concentration of Pdots and the irradiation density become inconsistent with the photocatalytic activity test. What’s the purpose of changing these factors and why is 2.7 mW/cm² determined as the irradiation density?

Response by the author: Thanks for the question. When we measured the external quantum yield of PFBT-T/PCBM Pdots, the basic principle was to see the theoretically highest performance. According to Beer’s law, the transmittance (T) of the incident light is expressed as:

$T=10^{-A}$, where A is the absorbance at the wavelength of incident light.

Therefore, we increased the concentration of Pdots to improve the light-harvesting efficiency. For the quantum yield test, we must use monochromatic light. For the 450 nm quantum yield test, we used 2.7 mW/cm² light intensity as it is similar to the light intensity of 450 nm in 1 Sunlight.

6. The pH value change before and after the photocatalytic reaction is important, but not considered.

Response by the author: Thank you for this comment, the pH before and after the reaction can help us understand the photocatalytic mechanism. After 5 h reaction, the pH value did not change much (from 7.4 to 7.6). As the pH change is small, we don’t think it will affect the overall photocatalytic reaction in our system during the test period.

7. Many spelling mistakes must be carefully checked and corrected throughout the manuscript, such as “graphitic” in line 38 and “framework” in line 39.

Response by the author: The manuscript has been checked thoroughly and some spelling mistakes have been corrected.

Reviewer #3 (Remarks to the Author):

This is an interesting paper, but mostly because of the tethered TEMPO catalyst they have developed. This may be interesting in catalysis beyond the specific H₂O₂ generation that is the subject of this communication. In the context of the title “Photocatalytic Hydrogen Peroxide production ...” I am not convinced this is practical for the production of hydrogen peroxide, particularly with EQE of 0.75% it does not qualify as “outstanding production rates”, as suggested on page 4. At the end of this inefficient process, one has a mixture of H₂O₂, methanol, formaldehyde and the catalyst-polymer. While the widely used anthraquinone method has its separation challenges, is there an improvement here?

Response by the author: Thank you for this comment, the word ‘outstanding’ has been replaced and the relative description has been changed in the revised manuscript:

‘The prepared binary PFBT-T/PCBM Pdots showed, compared to PFBT/PCBM Pdots, groundbreaking production rates of both H₂O₂ and formaldehyde of 865 μmol h⁻¹ g⁻¹_{Pdots} and an external quantum yield (EQE) of 0.75 % at 450 nm in neutral condition.’

Separation challenges do exist in both this work and ‘Anthraquinone method’ which is the industrial production method of H₂O₂. For the Anthraquinone method, metal catalysts such as Pd are required for the hydrogenation step where **temperature** and **pressure** need to be added. In the following oxidation step, oxygen will oxidize the hydroquinone to produce H₂O₂ and regenerate anthraquinone. During the hydrogenation reaction, by-products such as anthrone and dianthrone which are inert and **cannot be regenerated** will form. During the oxidation step, the oxidation by-product can be epoxides which **cannot lead to the formation of H₂O₂** (<https://www.ss-pub.org/wp-content/uploads/2014/11/BCR-E20140903-02.pdf>).

Therefore, firstly, the requirement of additional temperature and pressure is avoided in our work, which facilitates the sustainability process. Secondly, although we have ‘by-products’ such as formaldehyde, it is indeed a value-added product which can be used to make profits after separation. Based on the two points above, the method in our work is improved on both economic and sustainable aspects. However, we do agree that the separation of different products would be challenging and should be considered in future if the photocatalytic efficiency reaches an industrial application level.

The materials used have been well characterized, as has the progress of the reaction. In contrast a key reagent, light, is poorly characterized, as for the two light sources used (LED lamp and Xenon lamp with filter) no spectrum is shown. Further, the irradiation density (irradiance) is just a number given with no indication of how it was determined. This would make this work hard to reproduce. If this paper is resubmitted, here or elsewhere, this issue must be resolved.

Response by the author: Thank you for this comment. We have added the brands and spectra of LED and Xe lamps which we used for photocatalysis measurement in the revised SI as shown below:

'Photocatalytic activity test

Photocatalytic H₂O₂ generation reaction proceeded in a quartz cuvette. In detail, 2.5 mL Pdots reaction solution (22 μg mL⁻¹) which contains 5 M MeOH was purged by O₂ for 15 min to ensure the saturation of O₂ in solution (pH=7.4). Afterwards, the cuvette was irradiated by a LED lamp (Zenaro Lighting GmbH, SL-PAR38B/P17/50/E50/ND/27/UNI/EU/ZN, 420 - 750 nm, Spectrum shown in Figure S1) with an irradiation density of 50 mW cm⁻². The irradiation density was measured with a PM100D power meter (Thorlabs).

Quantum Yield Measurement

External quantum efficiency (EQE) of PFBT-T-PCBM Pdots was measured under the same condition as for photocatalytic H₂O₂ generation except that the concentration of Pdots was 90 μg mL⁻¹. The light source was replaced by Xenon lamp (CEL-HXF300, spectrum shown in Figure S2) with a light filter (QD 450 nm) and the irradiation density was 2.7 mW cm⁻². The irradiation density was measured with a PM100D power meter (Thorlabs).'

Figure S1. Spectrum of LED lamp used for photocatalysis.

Figure S2. Spectrum of Xe lamp that used for external quantum yield measurement. (provided by Beijing China Education Au-light Co., Ltd.)

REVIEWER COMMENTS

Reviewer #1 (Remarks to the Author):

The authors have thoroughly addressed my queries by conducting additional experiments, revising analysis methodologies, and adding further details to manuscript and SI. These additions have further improved the manuscript, and I have no further comments to add.

Reviewer #2 (Remarks to the Author):

In the revised manuscript, the authors have made some corrections accordingly. Certainly, their advance in Pdots systems should be encouraged, but I remain convinced the breakthrough is not outstanding enough to be published in the top journal Nature Communications:

1. Whether is the design strategy universal? In other words, how does it go when this design strategy is utilized in other systems, or replacing TEMPO with other molecules? Does PFBT-T/PCBM also show ground-breaking activity in other alcohol oxidation (e.g. ethanol, benzyl alcohol, etc.)? If so, I'd like to believe that this work can give a novel solution to a scientific question. However, there seems no sufficient evidence to persuade me in the present manuscript.

2. The authors claim that H₂O₂ generation goes through a two-step one-electron reduction of oxygen because electrons are less likely to accumulate in molecule catalysts. I wonder if PFBT-T/PCBM can produce only one electron in its excitation process. The trapping experiment, detection of intermediate (superoxide radicals) by ESR, and theoretical calculations such as free energy are not provided.

Reviewer #3 (Remarks to the Author):

The authors have responded satisfactorily to my comments. Frankly I was not over-enthusiastic about this paper, but clearly I was in a minority and I see that other reviewers recognized the value of this contribution.

As noted all my comments have been addressed and I am happy to see the improved version of the article.

Response Letter

Dear reviewers,

Thank you very much for reviewing our manuscript of "*Covalently linked organic molecular catalyst in conjugated organic polymer dots boosts photocatalytic hydrogen peroxide production and methanol oxidation in neutral condition*" (NCOMMS-23-58020A). All comments received are very helpful for us to improve the manuscript.

The manuscript now has been carefully revised according to all comments. The revised manuscript and supporting information with all changes highlighted in yellow are uploaded for review. Below we respond to the comments in detail.

Sincerely,

Haining Tian, on behalf of all co-authors

Reviewer #2

1. Whether is the design strategy universal? In other words, how does it go when this design strategy is utilized in other systems, or replacing TEMPO with other molecules? Does PFBT-T/PCBM also show ground-breaking activity in other alcohol oxidation (e.g. ethanol, benzyl alcohol, etc.)? If so, I'd like to believe that this work can give a novel solution to a scientific question. However, there seems no sufficient evidence to persuade me in the present manuscript.

Response by the authors: Thank you for this inspiring comment. TEMPO is an organic catalyst having excellent selectivity and efficiency of alcohol oxidation into aldehyde. It is the main purpose of this work to use it. Regarding other alcohol oxidation, we have tested PFBT-T/PCBM Pdots with ethanol and benzyl alcohol oxidation. PFBT-T/PCBM Pdots have shown reactivity to oxidize both substrates into corresponding aldehydes with H₂O₂ production as shown in the figure below. The experimental details can be found in the revised SI. Therefore, the strategy of TEMPO grafting is practical in various alcohol oxidations.

This figure has been added in SI (Figure S18) and the corresponding discussion has been provided as well.

2. The authors claim that H₂O₂ generation goes through a two-step one-electron reduction of oxygen because electrons are less likely to accumulate in molecule catalysts. I wonder if PFBT-T/PCBM can produce only one electron in its excitation process. The trapping experiment, detection of intermediate (superoxide radicals) by ESR, and theoretical calculations such as free energy are not provided.

Response by the authors: Thanks for this comment. It is hard to directly detect the superoxide radicals from EPR/ESR. It is common to use a trapping reagent to trap superoxide and then measure the EPR/ESR. However, considering the TEMPO radical that we already have in the system, it is challenging to get a clean EPR/ESR signal of the trapping reagent reacted with superoxide. Instead, we conducted a trapping experiment with benzoquinone (BQ) as the scavenger of superoxide radicals (*Environ. Sci. Pollut. Res. Int.* 2020, 27(25):31289-31299) and measure the formation of products. From the experimental results, one can see that a negligible amount of H₂O₂ was detected in the system containing 10 mM BQ although a similar amount of HCHO was generated. In order to rule out the direct electron transfer from Pdots to BQ completely competes the formation of superoxide, an additional control experiment has been done in presence of BQ, but in the absence of oxygen. In this case the result shows that HCHO is formed, but with much lower generation rate, only ca. 13% of the rates from the systems with O₂. This proves that the inhibited generation of H₂O₂ in presence of BQ and O₂ was dominantly from the consumption of the formed superoxide radicals by BQ, This result strongly suggests that the photocatalytic H₂O₂ production based on PFBT-T/PCBM is a two-step one-electron pathway based on the generation of superoxide radicals as the intermediate.

Table S2 has been added in the revised SI and the description has been added in the revised manuscript.

Table S2. Results of superoxide scavenger experiments

Benzoquinone (BQ)	O ₂	H ₂ O ₂ generation rate ($\mu\text{mol g}^{-1} \text{h}^{-1}$)	HCHO generation rate ($\mu\text{mol g}^{-1} \text{h}^{-1}$)
None	Saturated	865	844
10 mM	Saturated	0	797
10 mM	None	0	113

‘By adding benzoquinone (BQ) as the scavenger of superoxide radicals (O₂^{•-}), the H₂O₂ generation with PFBT-T/PCBM Pdots was completely inhibited while the formaldehyde production rate remained (Table S2), further demonstrating that the photocatalytic H₂O₂ should go through a two-step one-electron transfer pathway with O₂^{•-} as an intermediate.’

REVIEWERS' COMMENTS

Reviewer #2 (Remarks to the Author):

The concerns from the reviewer have been addressed, and thus this manuscript can be considered to be accepted for publication.